# Rapid screening of sixty potato cultivars for starch profiles to address a consumer glycemic dilemma

**Rocio Rivas[1], Edward Dratz[2], Thomas Wagner[3], Gary Secor[4], Amanda Leckband[1], David C. Sands[1]***

**1** Department of Plant Sciences and Plant Pathology, Montana State University, Bozeman, Montana, United States of America, **2** Department of Chemistry and Biochemistry, Montana State University, Bozeman, Montana, Montana, United States of America, **3** Tater Seed, Everett, Washington, United States of America, **4** Department of Plant Pathology, North Dakota State University, Fargo, North Dakota, United States of America

* davidsands41@yahoo.com

**Data Availability Statement:** All relevant data are within the manuscript and its Supporting information files. Supporting files contain

## Abstract

Potatoes are a dietary staple consumed by a significant portion of the world, providing valuable carbohydrates and vitamins. However, most commercially produced potatoes have a high content of highly branched amylopectin starch, which generally results in a high glycemic index (GI). Consumption of foods with high levels of amylopectin elicit a rapid spike in blood glucose levels, which is undesirable for individuals who are pre-diabetic, diabetic, or obese. Some cultivars of potatoes with lower amylopectin levels have previously been identified and are commercially available in niche markets in some countries, but they are relatively unavailable in the United States and Latin America. The high glycemic index of widely available potatoes presents a problematic "consumer's dilemma" for individuals and families that may not be able to afford a better-balanced, more favorable diet. Some native communities in the Andes (Bolivia, Chile, and Peru) reportedly embrace a tradition of providing low glycemic tubers to people with obesity or diabetes to help people mitigate what is now understood as the negative effects of high blood sugar and obesity. These cultivars are not widely available on a global market. This study examines 60 potato cultivars to identify potatoes with low amylopectin. Three independent analyses of potato starch were used: microscopic examination of granule structure, water absorption, and spectrophotometric analysis of iodine complexes to identify potato cultivars with low amylopectin Differences among cultivars tested were detected by all three types of analyses. The most promising cultivars are Huckleberry Gold, Muru, Multa, Green Mountain, and an October Blue x Colorado Rose cross. Further work is necessary to document the ability of these low amylopectin cultivars to reduce blood glucose spike levels in human subjects.

## Introduction

Current lifestyles have led much of the general population to become relatively sedentary and to adopt questionable eating habits, including high consumption of foods which appear to

information on where to locate the strains of potato.

**Funding:** This paper was supported by Western SARE Project SW12-108, Montana State University Agricultural Experiment Station.

**Competing interests:** The authors have declared that no competing interests exist.

drive an alarming rise in the incidence of obesity and Type 2 diabetes worldwide [1–3]. This problem is especially severe in low-income populations, where inexpensive high-energy foods tend to be consumed, instead of more expensive foods that have more balanced nutrition [4]. Potatoes are a popular staple food, available to consumers at a low cost per pound that can be easily stored. Potato starch is also widely used in the food industry as a processed food ingredient [5, 6]. Most varieties of potatoes have high levels of amylopectin starch, which is highly branched and rapidly digested. High levels of amylopectin cause a rapid spike in blood glucose level that occurs after consumption of high glycemic index potatoes [2, 7–9].

Fig 1 illustrates the difference in response of a typical individual's blood sugar levels to high and low glycemic index foods. Most varieties of potatoes are in the high glycemic category of foods that people with pre-diabetes, diabetes, or obesity should seek to avoid. Not only do high glycemic foods give a large spike in blood sugar levels, but the body's response to try to bring the level down tends to cause what is referred to as an undershoot, bringing levels below its desirable amount. This low level of blood sugar tends to stimulate hunger and the desire to consume more high glycemic foods. It follows that high glycemic foods tend to cause a "roller coaster" effect on blood sugar and may cause excessive food consumption [10].

The high glycemic index of widely available potatoes presents a problematic "consumer's dilemma" for individuals and families that may not be able to afford a better-balanced, more favorable diet [2, 7–9]. The food industry in some countries have addressed the high glycemic index potato problem particularly in Denmark and Australia [8, 11], by providing and promoting lower glycemic varieties of potatoes that are relatively unavailable in the United States and most other countries. Some native communities in the Andes (Bolivia, Chile, and Peru) reportedly embrace a tradition of providing low glycemic tubers to people with obesity or diabetes [12].

Potato starch consists of two major types of carbohydrates, amylose, and amylopectin [13]. Amylose, less abundant in most varieties of potato tubers, has a linear repeating glucose 1–4 linked polysaccharide backbone structure that digests relatively slowly. Amylopectin shares the glucose 1–4 linked chains and additionally contains highly branched beta 1–6 side chains that produce a broader structure [8, 14–20], that digests much faster than amylose [8, 14, 15, 17–19] that can present metabolic stress. Starch granule synthesis proceeds with the formation of amylose first and amylopectin branches are layered on the outside of the granules [22]. Typical high amylopectin potato starch granules have a smooth rounded surface with concentric rings, formed by amylopectin [21, 22].

Researchers have found that a higher ratio of amylose/amylopectin correlates with loss of surface smoothness in starchy granules [9, 23]. We noticed when screening potato starch granules, that some potato cultivars showed a highly disrupted starch granule surface, observed under the bright field microscope at 400x, while granules from other cultivars have a smooth appearance. Thus, the surface structure of the starch granules might be useful as an initial screen for identifying promising varieties with lower amylopectin starch content, that would favor a low Glycemic Index (GI) [9, 16, 20, 24, 25]. We test this possibility by also testing the swelling capacity (water absorption capacity) of the starch of the same varieties, which has also been reported to correlate with the amylose/amylopectin ratio [18, 25–29]. Finally, we used an iodine inclusion spectrophotometric method that allows quantification of amylose and amylopectin in mixtures [16, 30, 31]

Zhu et al. [32], observed that high amylose rice consumption had beneficial health effects on obese rats, by reducing blood sugar levels and body weight, which would presumably also have positive health benefits for humans [2, 23, 32, 33]. This might be consistent with beliefs and dietary recommendations about types of plants that are favorable among Native communities in Peru and Chile, but unfortunately most of their traditional potato germplasm was lost

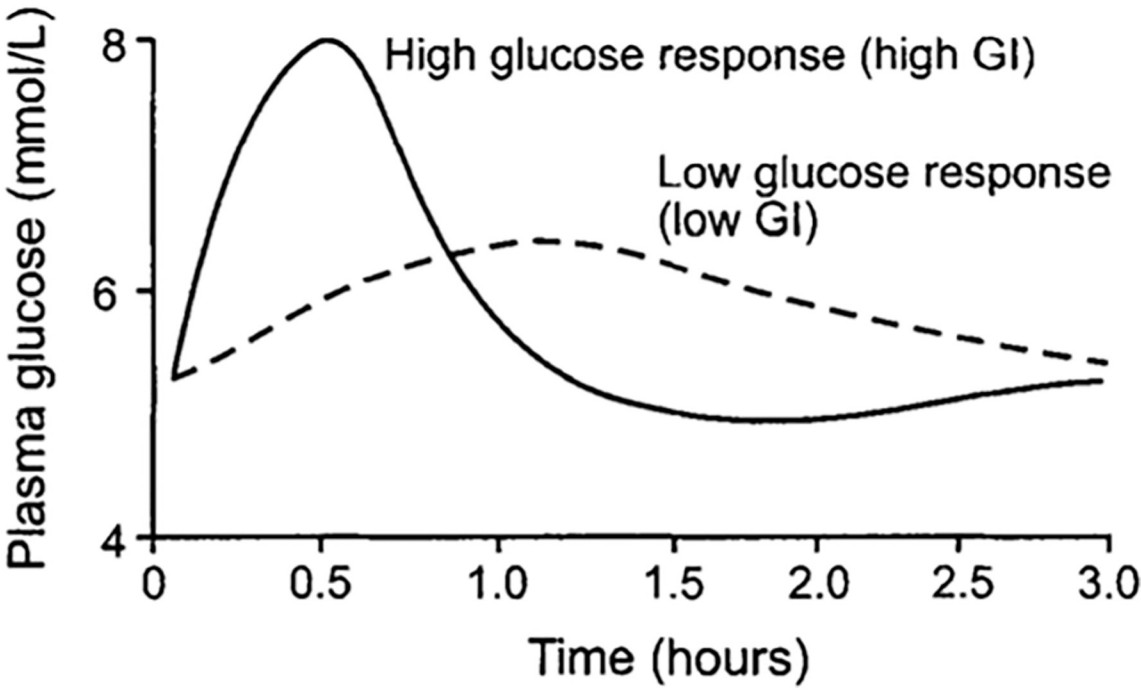

**Fig 1. Typical blood plasma glucose response (mmol/l) from a high compared to a low glycemic index (GI) food.** The high glycemic food tends to cause an "undershoot" in the blood glucose level, below the desirable level, which also tends to stimulate hunger and consumption of additional high glycemic foods [10].

during the colonization of the region [12]. However, potato varieties maintained in seed banks or that may already have been commercialized might possess beneficial, low glycemic traits. Therefore, we screened 60 potato varieties and selections in this study to investigate the relationship between starch granule conformation, swelling power and amylose: amylopectin ratios, seeking to validate rapid screening methods for low GI potential. The favorable lines revealed from this screening study can be used for further agronomic, breeding, and human GI testing.

## Methods and materials

### Potato material tested

A collection of 60 potato varieties and selections of potato germplasm were obtained from research laboratories of Montana State University, the USDA potato collection [34], commercial seed producers, and private breeder Tom Wagner, Everett, Washington. These varieties were tested for low GI potential using starch granule morphology, water absorption capacity and spectrophotometric analysis of amylose and amylopectin content, as described below. The potato materials were also chosen for their favorable agronomic traits and were increased in the MSU greenhouse for testing. Three tubers of each variety or selection were analyzed as depicted in Fig 2.

### Morphological analysis of starch granules

Tubers were assayed several months after harvest. A cork-borer was used to remove a core (1cm x 1 cm) from the center medullary tissue of tubers for observation. A subsample 5 x 5

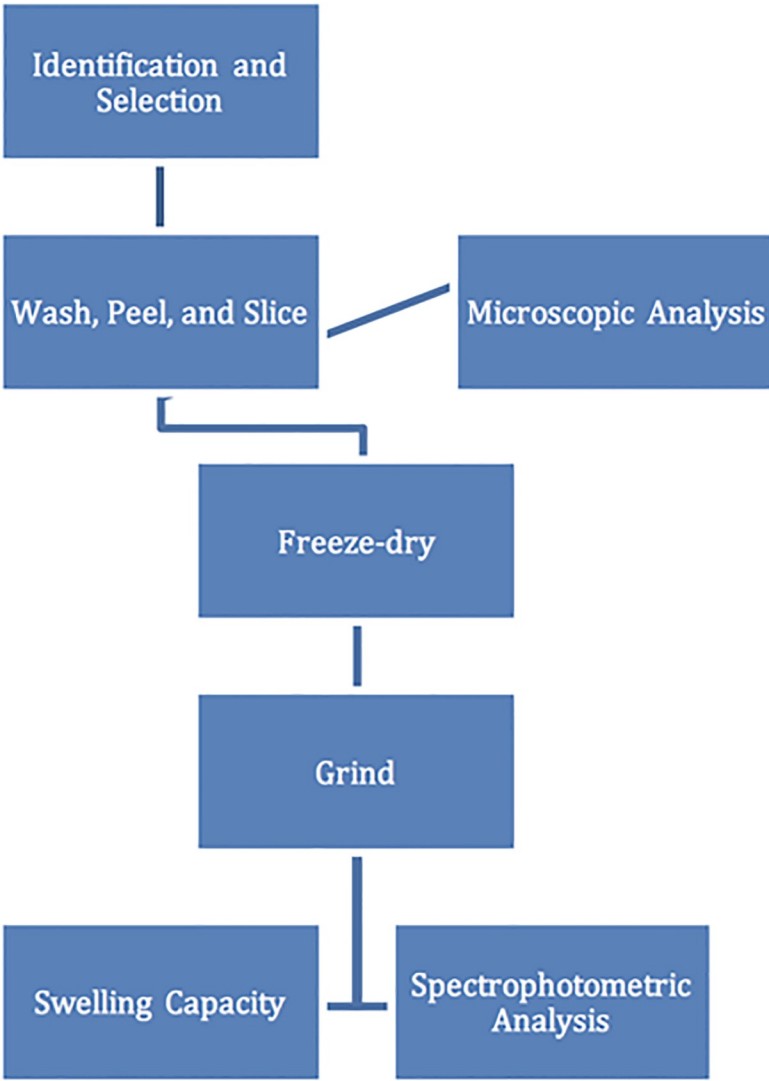

**Fig 2. Flow chart illustrating starch analysis conducted on 60 selected potato varieties and selections.**

mm$^2$ was pressed onto a microscope slide, with a gloved finger, covered with a cover slip, viewed with a bright field microscope (Nikon Eclipse E440) at 400x magnification and photographed. Starch granules from each variety or selection was numerically graded, using a grading scale assessment (GSA). The GSA was graded according to the numerical scale described in Fig 3. Samples of three different tubers of each variety or selection were examined and photographed. Each variety was given more than one rating score if the granules within the picture showed diversity, as explained further in the results section.

### Water absorption capacity of starch

The swelling capacity (SC) or water absorption capacity (WAC) measures the ratio between the weight of the hydrated starch flour gel and the dry flour samples. Samples with increased amylose concentrations support lower water absorption after starch jellification [25–27, 35–37]. WAC was evaluated using a procedure modified from Martin [36]. Potato samples were

| Description | Whole smooth granule, halo visible | Whole smooth granule with points | Whole smooth granule with wrinkles | Whole granule with surface alteration | Granule with fracturing | Collapsed granule |
|---|---|---|---|---|---|---|
| Images | | | | | | |
| Ranking | 0 | 1 | 2 | 3 | 4 | 5 |

**Fig 3.** The grading scale used for starch granule assessment: 0 = whole granules with smooth visible hilum 1 =. Whole smooth granules with dots 2 =. Whole smooth granules with surface alterations (wrinkles) 3 =. Whole granules with large wrinkles 4 =. Starch granule surface fractures 5 = collapsed granule.

freeze dried and processed into powdered form in a coffee grinder to obtain potato flour. Thirty to forty mg of potato flour was weighed in pre-weighed 2 mL screw top glass tubes and the total weight measured to four significant figures. To each tube 1.5 ml of distilled water was added to the tubes, which were capped, vortexed, placed in a thermo-mixer (Thermo Fisher) at 92 ˚C. The tubes were mixed at 800 RPM for 30 minutes and centrifuged at 1000g for 10 minutes at 4˚ C. The supernatant was aspirated from the tubes, carefully avoiding the gelatinous layer, and the tubes reweighed. The capacity to absorb water was calculated by dividing the flour plus water weight by the dry flour weight.

## Spectrophotometric analysis of starch properties

Rundle [30] first reported a marked difference in iodine inclusion and absorption spectra between amylose and amylopectin and developed a spectrophotometric method that allows quantification of both components in mixtures. We used this spectrophotometric method, following modifications by Jarvis and Walker [31] and Fajardo, et al. [16], to quantify the amylose/amylopectin ratios. 20 mg of freeze-dried potato flour was suspended in 2ml of 80% ethanol in a 15 ml plastic tube and briefly mixed with a Vortex Shaker to suspend the flour. Samples were centrifuged at room temperature for 60 seconds at top speed in a clinical centrifuge and the supernatant carefully removed. This washing procedure was repeated twice with the same solvent. The resulting washed starch (pellet) was mixed with 5ml of water and 5ml of 1 M KOH in the same tube and vortexed. It is important to use ultra-pure water in the analysis for reproducible results. Also, the absorbance value of the iodine solution at 550 nm versus solvent should be near 0.1, as the iodine solution is light and age sensitive, which will increase the absorbance [16]. Precision in the preparation of the iodine and the amylopectin/amylose solutions is crucial to obtaining reliable/reproducible amylose and amylopectin determinations. Since we are determining the proportion of amylose content rather than the exact amount of amylose in the potato starch, there is no purification step needed for the isolation of pure starch from the potato powder as used by other determination methods. One ml of this mixture was transferred into a 50 ml tube, neutralized with 5ml of 0.1M HCl, mixed with 0.5 ml of Lugol solution [31], and adjusted to 50ml final volume with distilled water. The samples were prepared in triplicate and allowed to stand for 15 min at room temperature, before

measurement to stabilize the inclusion of the iodine. Each sample was placed in a one cm plastic spectrophotometer cuvette and absorption spectra recorded from 480–800 nm, versus a distilled water blank, using a Genesis 10 UV Instrument (Thermo-Fisher Scientific). The spectra were analyzed using the Vision Lite software (Thermo-Fisher). The data obtained was processed using the statistical software Stat Plus: Mac LE 2009.

## Results

### Morphological analysis of starch granules

We noticed that for most potato varieties and selections, virtually all the granules in each sample were quite similar in appearance. In other cases, there was a high diversity of granular appearance. For these cases, two scores were assigned, as shown in Table 1 with a rough percentage of the granules the scores represented. Fig 4 shows that the granules in the common commercial varieties Russet Burbank and Yukon Gold, known to have a high amylopectin content (72% and 69% respectively), showed very smooth granules, with a distribution in the sizes of their granules. The previously shown grading scale in Fig 3 dictated that the granules in Fig 4 fit a GSA rating of 1.

In other potato samples, there was considerable diversity of granular appearance. For these cases, two scores were assigned, as shown in Table 1 with a rough percentage of the granules each of the scores represented. The previously shown grading scale in Fig 3 dictated that the granules in Fig 4 fit a GSA rating of 1.

In contrast, the starch granules in only a few of the potato varieties showed significant rough, granular surfaces, and thus had much higher GSA values. Huckleberry Gold, Muru, and Mich-Oct have GSA ratings of 5, as shown in Fig 5, where 90–95% of the total granules observed presented collapsed surfaces, the other 5–10% correspond to granules with minor irregularities. The variety Multa graded 4 GSA with 60–65% of the granules presenting a surface fracture level, and the 35–40% corresponded to minor irregularities.

The GSA ratings and percent starch granules in the GSA rating of 60 varieties and selections are listed in Table 1.

### Water absorption capacity of starch

During the development of the assay to measure the water absorption capacity (WAC), we found that it was essential to handle the samples very carefully after the final centrifugation, with minimum vibration, since complete removal of the supernatant was difficult in some samples. The WAC data for 60 varieties and selections, is shown with standard deviations in Table 2. The ten varieties with the lowest water absorption were the same as those with the highest GSA scores. These included Huckleberry Gold, Muru, Bzura, Michigan October, and the F-1 hybrid Michigan October X Colorado Rose (Fig 2). The October Blue x Colorado Rose hybrid was ranked third lowest in its relative capacity to absorb water.

### Determination of amylose percentage

We encountered a wide range of variation in spectrophotometric evaluation when we assessed amylose percentage, using the Jarvis and Walker approach [31]. Fajardo et al. [16], in contrast, reported that using absorbance changes at the optimum wavelengths can provide highly consistent results. This can be seen in Fig 6. Fajaro, et al. [16] found that the maximum absorbance of the amylose-iodine complex was at 620 nm and the maximum absorbance of the amylopectin-iodine complex was at 550 nm. Thus, the percent amylose can be determined from

**Table 1. GSA rating and range of uniformity of starch granules ratings from the 60 potato varieties and selections tested.**

| Variety or Selection | GSA Rating | % Uniformity |
|---|---|---|
| Bjorna | 5 | 55–60 |
| Bzura | 5 | 70–75 |
| Huckleberry Gold | 5 | 100 |
| I 1035 | 5 | 90–95 |
| Laram K'anchali | 5 | 75–80 |
| Muru | 5 | 100 |
| Olalla | 5 | 75–80 |
| Arma | 4 | 45–50 |
| Bolivian Blizzard | 4 | 80–85 |
| Bareroot River | 4 | 85–90 |
| I 1036 | 4 | 85–90 |
| Multa | 4 | 60–65 |
| Green Mountain | 3 | 60–65 |
| Iker | 3 | 45–50 |
| Juice Valley | 3 | 45–50 |
| Lumper | 3 | 50–55 |
| Marble Gold | 3 | 85–90 |
| Monona | 3 | 35–40 |
| Oct Blue | 3 | 85–90 |
| Oct Blue x Col Rose | 3 | 85–90 |
| Pirola | 3 | 65–70 |
| Biddy Taro | 2 | 65–70 |
| Bison | 2 | 90–95 |
| C97007 | 2 | 85–90 |
| Cherry red | 2 | 80–85 |
| Katelidan | 2 | 45–50 |
| Norland | 2 | 75–80 |
| Papa Amorga | 2 | 85–90 |
| Purple Valley | 2 | 65–70 |
| Red Pontiac | 2 | 50–55 |
| Allegany | 1 | 20–30 |
| Anolla | 1 | 35–40 |
| Asun | 1 | 45–50 |
| Bolivian Spring | 1 | 90–95 |
| Russet Burbank | 1 | 90–95 |
| Charlotte | 1 | 90–95 |
| Chella x Bulk Clover | 1 | 90–95 |
| Garnet Chile | 1 | 90–95 |
| Golden Anniversary | 1 | 85–90 |
| Goldra | 1 | 80–85 |
| Leona | 1 | 85–90 |
| Mich Oct x John T | 1 | 85–90 |
| NH x SPG | 1 | 45–50 |
| Nicola | 1 | 45–50 |
| Phytophyter | 1 | 35–40 |
| Picasso | 1 | 45–50 |

*(Continued)*

**Table 1.** (Continued)

| Variety or Selection | GSA Rating | % Uniformity |
|---|---|---|
| Rush Share | 1 | 45–50 |
| Sangre | 1 | 45–50 |
| Sassy Lassy | 1 | 85–90 |
| Skagit Magic | 1 | 90–95 |
| Violet Butter | 1 | 40–45 |
| X gem | 1 | 60–75 |
| Yukon Gold | 1 | 80–85 |
| Arcilla-597779 | 0 | 100 |
| Chelan | 0 | 100 |
| Chipitiquilla | 0 | 100 |
| Dark Red Norland | 0 | 100 |
| Fontenay | 0 | 100 |
| Rose Valley | 0 | 100 |
| Studebaker x Nordic Oct | 0 | 100 |

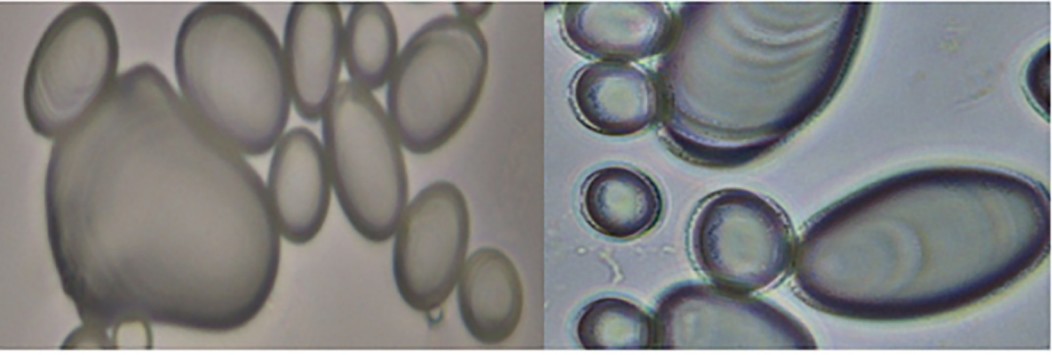

**Fig 4. Bright field microscopic photos of starch granules of Yukon Gold (left) and Russet Burbank (right) at 400x.** These granules were given a GSA rating of 1, according to the Grading Scale shown in Fig 3.

| Type | Huckleberry | Muru | Multa | Bzura | Mich. October |
|---|---|---|---|---|---|
| Image | | | | | |

**Fig 5. Starch granule images from left to right are Huckleberry Gold, Muru, Multa, Bzura, and Michigan October.**

**Table 2. Water absorption capacity (WAC) and standard deviations of 60 potato varieties and selections.**

| Variety | Water Absorption Capacity WAC | | Standard Deviation |
|---|---|---|---|
| Monona | 21.6 | ± | 2.3 |
| Chipitiquilla | 21.5 | ± | |
| Skagit Magic | 20.8 | ± | 2.3 |
| Picasso | 20.5 | ± | |
| Cherry red | 20.3 | ± | 1.7 |
| C97007 | 20 | ± | 1.2 |
| Katelidan | 19.4 | ± | 3.9 |
| Purple Valley | 18.3 | ± | 4.5 |
| Bison | 18.2 | ± | 2.3 |
| Rose Valley | 18 | ± | 1.6 |
| Garnet Chili | 17.8 | ± | 2.2 |
| Chelan | 17.4 | ± | |
| Nicola | 17.3 | ± | 2.1 |
| Iker | 17.2 | ± | 1.6 |
| Phytophyter | 17 | ± | 2.1 |
| Dark Red Norland | 16.9 | ± | 4.1 |
| Juice Valley | 16.4 | ± | 2.4 |
| Bolivian Spring | 16.1 | ± | 3.7 |
| Charlotte | 15.8 | ± | 2.8 |
| Golden Anniversary | 15.7 | ± | 2.8 |
| Goldra | 15.7 | ± | 2.7 |
| Pirola | 15.7 | ± | 0.9 |
| Asun | 15.6 | ± | 3 |
| Anolla | 15.5 | ± | 0.9 |
| Laram K'anchali | 15.5 | ± | 1.2 |
| I 1035 | 15.4 | ± | 5 |
| Bjorna | 15.4 | ± | 2.1 |
| Sassy Lassy | 15.2 | ± | 2.3 |
| Chella x Bulk Clover | 15.1 | ± | 1.1 |
| NH x SPG | 14.9 | ± | 1.2 |
| Sangre | 14.9 | ± | 0.5 |
| Violet Butter | 14.7 | ± | 1.3 |
| Arma | 14.5 | ± | 2.6 |
| I 1036 | 14 | ± | 3.1 |
| Studebaker x Nordic Oct | 13.8 | ± | 1.9 |
| Bzura | 13.7 | | |
| Rush Share | 13.6 | ± | 1.1 |
| X Gem | 13.6 | ± | 0.8 |
| Leona | 13.4 | ± | 1 |
| Bolivian Blizzard | 13.2 | ± | 1.7 |
| Olalla | 13.2 | ± | 1.6 |
| Bareroot River | 13.1 | ± | 3.2 |
| Arcilla-597779 | 12.9 | ± | 1.2 |
| Allegany | 12.9 | ± | 1.9 |
| Green Mountain | 12.8 | ± | 1.3 |
| Lumper | 12.7 | ± | 1.3 |
| Multa | 12.3 | ± | 1.8 |

(*Continued*)

**Table 2.** (Continued)

| Variety | Water Absorption Capacity WAC | | Standard Deviation |
|---|---|---|---|
| Biddy Taro | 12.2 | ± | 0.6 |
| Fontenay | 12.1 | ± | 1.3 |
| Marble Gold | 12 | ± | 1.1 |
| Oct Blue | 12 | ± | |
| Norland | 11.5 | ± | 0.8 |
| Yukon Gold | 11.2 | ± | 2.1 |
| Red Pontiac | 10.9 | ± | 2 |
| Papa Amorga | 10.6 | ± | 0.7 |
| Muru | 10.5 | ± | 0.6 |
| Mich Oct x John T | 10.4 | ± | 0.3 |
| Oct Blue x Col Rose | 9.9 | ± | 0.9 |
| Huckleberry Gold | 9.5 | ± | 0.7 |
| Russet Burbank | 9.5 | ± | 0.8 |

absorbances at the peak wavelengths of each species as follows [16]:

$$\text{Amylose \%} = y = 107.7\,x - 77.4$$

Where x = A620nm/A550nm, measured against the reagent blank.

The amylose ratings of 60 potato varieties and selections are shown in Table 3. We screened the 60 potato varieties in triplicate to provide the standard deviations in Table 3. The potato varieties with a higher concentration of amylose are Green Mountain, Muru, Bzura, Multa, Huckleberry Gold, and the F-1 cross October Blue X Colorado Rose. These samples also had high surface alterations of their starch granules, which supports our initial hypothesis that surface roughness was due to higher amylose content.

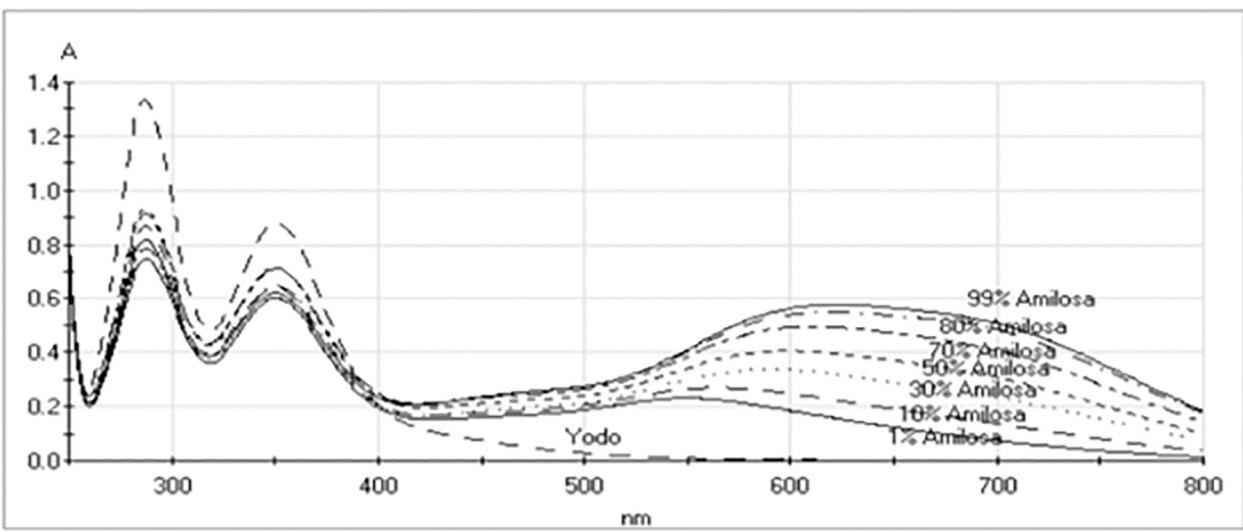

**Fig 6. Spectra of solutions of known concentrations of potato starch polysaccharide mixtures of amylose (amilosa) and amylopectin.** The spectrum marked Yodo is the iodine containing reagent solution blank.

**Table 3. Amylose content (%) of 60 potato varieties and selections.**

| Variety | Amylose (%) |
| --- | --- |
| Green Mountain | 60.9 |
| Muru | 59.0 |
| Bzura | 50.4 |
| Multa | 44.7 |
| Huckleberry Gold | 44.7 |
| Oct Blue x Col Rose | 43.8 |
| Laram K'anchali | 43.8 |
| NH x SPG | 40.4 |
| Lumper | 39.7 |
| Anolla | 39.5 |
| Bjorna | 38.3 |
| Iker | 37.1 |
| Sassy Lassy | 37.1 |
| X Gem | 36.6 |
| Chipitiquilla | 35.7 |
| Red Pontiac | 35.0 |
| Bolivian Blizzard | 34.7 |
| Phytophyter | 34.5 |
| Papa Amorga | 34.2 |
| Skagit Magic | 34.2 |
| Oct Blue | 33.6 |
| Dark Red Norland | 33.3 |
| Juice Valley | 33.3 |
| Studebaker x Nordic Oct | 32.9 |
| Cherry Red | 32.8 |
| Marble Gold | 32.7 |
| Fontenay | 32.5 |
| Nicola | 32.3 |
| Leona | 32.3 |
| Arma | 31.9 |
| Mich Oct x John T | 31.8 |
| Arcilla-597779 | 31.4 |
| Pirola | 31.2 |
| Rose Valley | 31.2 |
| Violet Butter | 30.9 |
| Norland | 30.8 |
| Bolivian Spring | 30.7 |
| Katelidan | 30.7 |
| I 1035 | 30.4 |
| I 1036 | 30.4 |
| Yukon Gold | 30.4 |
| Olalla | 30.0 |
| Biddy Taro | 29.8 |
| Bareroot River | 29.2 |
| Sangre | 28.9 |
| Goldra | 28.8 |
| Garnet Chile | 28.3 |

(*Continued*)

**Table 3.** (Continued)

| Variety | Amylose (%) |
|---|---|
| Asun | 27.7 |
| Monona | 27.6 |
| C97007 | 27.4 |
| Charlotte | 27.1 |
| Chelan | 26.6 |
| Rush Share | 26.6 |
| Bison | 26.2 |
| Golden Anniversary | 25.0 |
| Allegany | 24.1 |
| Chella x Bulk Clover | 24.0 |
| Picasso | 23.9 |
| Purple Valley | 23.9 |
| Russet Burbank | 23,7 |

## Calculation of potential glycemic index based on amylose percentage

Moreira's study in 2012 [38] measured the blood sugar levels in subjects two hours after eating a starchy meal with a known concentration of amylose and reported a correlation equation where x is the amylose percentage of total starch content and GIp is the GI potential value (GIp). Using this relationship and the amylose content of potato varieties and selections, a GIp can be calculated (Table 4).

The GIp for Green Mountain and Muru, the two varieties with the highest percentage of amylose have negative values, which suggests that such high amylose samples were outside the range of Moreira's equation.

Pearson's correlation coefficient tests were performed to study the possible relationship between granular surface appearance of starch granules (GSA), granule absorption capacity (GAC), and amylose content by spectrophotometric analysis in multiple potato varieties and selections. There was a negative correlation between GAC and amylose content $r^2 = -0.34$, $p > 0.01$), confirming the hypothesis that amylose inhibits water absorption by potato starch. There was a positive correlation between GSA scores and the amylose content of the potato starch ($r^2 = 0.46$, $p < 0.05$). These comparisons can be seen in Table 5. These results agree with other observations with potatoes [39] and peas, as high amylose peas are more wrinkled and cause a much lower insulin response in humans than smooth peas [40]. The wrinkled peas also have a lower capacity to absorb water [41].

## Discussion

Nutrition data indicates that consumption of potatoes with higher concentrations of amylose is expected to have a lower Glycemic Index and would be expected to provide health benefits for people that have prediabetes, diabetes and/or obesity. A lower sugar release during digestion due to differences of digestibility of amylose and amylopectin is expected to cause a lower sugar spike and lower release of insulin [4, 12, 32, 40, 42]. Furthermore, high insulin release tends to lead to an undershoot in desirable blood sugar levels (hypoglycemia), stimulating hunger, increased food consumption and potential weight gain, as illustrated in Fig 1.

By testing 60 potato cultivars, we have been able to identify the six most promising candidate cultivars for further agronomic evaluation for consumer acceptability. Those samples are

**Table 4. Predicted glycemic index (GIP) of 52 potato varieties and selections calculated using the equation described by Moreira [38].** GIp = Glycemic Index potential.

| Variety | % amylose | GI potential | Variety | % amylose | GI potential |
|---|---|---|---|---|---|
| Green Mountain | 60.9 | (14.69) | Mich Oct x John T | 31.8 | 73.28 |
| Muru | 59.0 | (8.94) | Arcilla-597779 | 31.4 | 74.43 |
| Bzura | 50.4 | 16.93 | Pirola | 31.2 | 75.14 |
| Huckleberry | 44.7 | 34.18 | Rose Valley | 31.2 | 75.14 |
| Multa | 44.7 | 34.18 | Violet Butter | 30.9 | 75.90 |
| Oct Blue x Col Rose | 43.8 | 37.06 | Norland | 30.8 | 76.17 |
| NH x SPG | 40.4 | 47.12 | Bolivian Spring | 30.7 | 76.65 |
| Lumper | 39.7 | 49.35 | Katelidan | 30.7 | 76.66 |
| Anolla | 39.5 | 49.99 | I 1035 | 30.4 | 77.31 |
| Bjorna | 38.3 | 53.57 | Yukon Gem | 30.4 | 77.31 |
| Iker | 37.1 | 57.15 | I 1036 | 30.4 | 77.51 |
| Sassy Lassy | 37.1 | 57.22 | Olalla | 30.0 | 78.72 |
| X gem | 36.6 | 58.62 | Biddy Taro | 29.8 | 79.28 |
| Chipitiquilla | 35.7 | 61.45 | Bareroot River | 29.2 | 81.13 |
| Red Pontiac | 35.0 | 63.54 | Sangre | 28.9 | 81.99 |
| Bolivian Blizzard | 34.7 | 64.46 | Goldra | 28.8 | 82.25 |
| Phytophyter | 34.5 | 65.17 | Garnet Chile | 28.3 | 83.65 |
| laram K'anchali | 34.2 | 65.81 | Asun | 27.7 | 85.70 |
| Skagit Magic | 34.2 | 65.81 | Monona | 27.6 | 85.73 |
| Papa Amorga | 34.2 | 65.96 | C97007 | 27.4 | 86.43 |
| Oct Blue | 33.6 | 67.76 | Charlotte | 27.1 | 87.37 |
| Dark Red Norland | 33.3 | 68.68 | Chelan | 26.6 | 88.81 |
| Juice Valley | 33.3 | 68.75 | Rush Share | 26.6 | 88.89 |
| Studebaker x Nordic Oct | 32.9 | 69.77 | Bison | 26.2 | 89.97 |
| Cherry Red | 32.8 | 70.12 | Golden Anniversary | 25.0 | 93.81 |
| Marble Gold | 32.7 | 70.58 | Allegany | 24.1 | 96.44 |

as follows: Huckleberry Gold, Muru, Bzura, October Blue x Colorado Rose, Multa, and Green Mountain. In addition to the consumer dilemma on choices of affordable healthy foods, there is the Breeders' Dilemma, that is the conflict between improved food nutrition that often reduces agronomic yield. However, agronomic yield usually drives the choice of varieties grown [43]. To pursue our efforts to identify nutritionally favorable potato varieties, we

**Table 5. Compilation of traits of seven selected cultivars with the most favorable starch content, compared to three cultivars with high amylopectin content.**

| Variety | % Amylose Content | Starch Granule Ranking | Water Absorption |
|---|---|---|---|
| Green Mountain | 61 | 3 | 13 |
| Muru | 59 | 5 | 10 |
| Bzura | 50 | 5 | 13 |
| Huckleberry Gold | 45 | 5 | 9 |
| Multa | 45 | 5 | 12 |
| Oct. Blue x Col Red | 44 | 3 | 9 |
| Monona | 28 | 2 | 22 |
| Picasso | 24 | 1 | 21 |
| Purple Valley | 24 | 2 | 18 |

identified patterns in potato starch properties that could be used in rapid screening. By screening of 60 potato lines, patterns were identified in the microscopic appearance of starch granules, confirmed by direct starch content assays, that permits rapid initial screening of candidate potato varieties for increased amylose contain and potentially lower Glycemic Index. This rapid screening system will potentially foster elevation of nutrition as a priority in plant breeding. This work now will be followed by glycemic index measurements and agronomic fitness [40, 43, 44] on the most promising potato varieties.

## Supporting information

**S1 File.**
(DOCX)

## Author Contributions

**Conceptualization:** Edward Dratz, David C. Sands.

**Formal analysis:** Edward Dratz, David C. Sands.

**Funding acquisition:** David C. Sands.

**Investigation:** Rocio Rivas, Thomas Wagner, Gary Secor, David C. Sands.

**Project administration:** David C. Sands.

**Resources:** Thomas Wagner.

**Supervision:** David C. Sands.

**Validation:** Gary Secor.

**Visualization:** David C. Sands.

**Writing – original draft:** Rocio Rivas, Edward Dratz, Gary Secor, Amanda Leckband, David C. Sands.

**Writing – review & editing:** Edward Dratz, Gary Secor, Amanda Leckband, David C. Sands.

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
