## [Decision Letter · Decision Letter 0]

11 Aug 2021

PONE-D-21-22200

Rapid screening methods of potato cultivars for low glycemic traits

PLOS ONE

Dear Dr. Sands,

Thank you for submitting your manuscript to PLOS ONE. After careful consideration, we feel that it has merit but does not fully meet PLOS ONE’s publication criteria as it currently stands. Therefore, we invite you to submit a revised version of the manuscript that addresses the points raised during the review process.

We look forward to receiving your revised manuscript.

Kind regards,

Umakanta Sarker

Academic Editor

PLOS ONE

Journal Requirements:

2. In order to enhance reproducibility, we ask that you provide within the manuscript or supplemental further details, such as the vendor, variety etc. of the samples used in the study.

Reviewers' comments:

Reviewer's Responses to Questions

**Comments to the Author**

1. Is the manuscript technically sound, and do the data support the conclusions?

Reviewer #1: Partly

Reviewer #2: Yes

2. Has the statistical analysis been performed appropriately and rigorously? 

Reviewer #1: No

Reviewer #2: Yes

3. Have the authors made all data underlying the findings in their manuscript fully available?

Reviewer #1: Yes

Reviewer #2: Yes

4. Is the manuscript presented in an intelligible fashion and written in standard English?

Reviewer #1: Yes

Reviewer #2: Yes

5. Review Comments to the Author

Reviewer #1: 1. Alpha amylase inhibitory activity (AAIA) of potatoes might have a significant role in lowering Glycemic Index (Low GI) in potatoes so AAIA related enzymatic data should be added to validate the result.

2. All tested potatoes have standard deviation data except Chipitiquilla, Picasso, Chelan, Bzura, Oct Blue. It needs to be explained.

3. In amylose (%) and potential glycemic index equations, R2 value should be mentioned with equation obtained.

4. More explanations are needed regarding your calculation of potential glycemic index. You are referring Moreira’s study in 2012 only but need to clarify the adopted equation, logic, condition and justification as you did not conduct any in vivo experiments on GI (AUC method).

5. When denoting correlation between two variables, you have to express in r (Small letter) not r2. (GSC score & Amylose and GAC & Amylose). For conducting Pearson’s correlation coefficient tests what is your level of significance?

Reviewer #2: Duplicate (or redundant) publication is the publication of a paper that is substantially similar to a published paper by the same author, without acknowledging the source and without obtaining the permission of the original copyright holder. There may be superfluous differences between the original and the second paper, such as a new title or a modified abstract, but the data set and findings stay the same.Researchers have a duty to make publicly available the results of their research on human subjects and are accountable for the completeness and accuracy of their reports. All parties should adhere to accepted guidelines for ethical reporting. Negative and inconclusive as well as positive results must be published or otherwise made publicly available. Sources of funding, institutional affiliations and conflicts of interest must be declared in the publication.

6. PLOS authors have the option to publish the peer review history of their article (what does this mean?). If published, this will include your full peer review and any attached files.

Reviewer #1: **Yes: **Dr. Habibul Bari Shozib

Reviewer #2: No

---

## [Author Response · Author response to Decision Letter 0]

21 Apr 2022

Attached is the edited manuscript

---

## [Decision Letter · Decision Letter 1]

9 May 2022

Rapid screening methods of potato cultivars for low glycemic traits

PONE-D-21-22200R1

Dear Dr. Sands,

We’re pleased to inform you that your manuscript has been judged scientifically suitable for publication and will be formally accepted for publication once it meets all outstanding technical requirements.

Kind regards,

Umakanta Sarker

Academic Editor

PLOS ONE

Additional Editor Comments (optional):

Reviewers' comments:

Reviewer's Responses to Questions

**Comments to the Author**

1. If the authors have adequately addressed your comments raised in a previous round of review and you feel that this manuscript is now acceptable for publication, you may indicate that here to bypass the “Comments to the Author” section, enter your conflict of interest statement in the “Confidential to Editor” section, and submit your "Accept" recommendation.

Reviewer #1: (No Response)

2. Is the manuscript technically sound, and do the data support the conclusions?

Reviewer #1: Yes

3. Has the statistical analysis been performed appropriately and rigorously? 

Reviewer #1: Yes

4. Have the authors made all data underlying the findings in their manuscript fully available?

Reviewer #1: Yes

5. Is the manuscript presented in an intelligible fashion and written in standard English?

Reviewer #1: Yes

6. Review Comments to the Author

Reviewer #1: (No Response)

7. PLOS authors have the option to publish the peer review history of their article (what does this mean?). If published, this will include your full peer review and any attached files.

Reviewer #1: **Yes: **Habibul Bari Shozib

---

## [Editor Report · Acceptance letter]

3 Feb 2023

PONE-D-21-22200R1 

Rapid Screening of Sixty Potato Cultivars for Starch Profiles to Address a Consumer Glycemic Dilemma 

Dear Dr. Sands:

I'm pleased to inform you that your manuscript has been deemed suitable for publication in PLOS ONE. Congratulations! Your manuscript is now with our production department. 

Kind regards, 

on behalf of

Professor Umakanta Sarker 

Academic Editor

PLOS ONE